# An agent-based model of the spread of behavioural risk-factors for cardiovascular disease in city-scale populations

**James Archbold[1]☯, Sophie Clohessy[2], Deshani Herath[2], Nathan Griffiths[1]☯\*, Oyinlola Oyebode[3]☯**

**1** Department of Computer Science, University of Warwick, Coventry, United Kingdom, **2** Warwick Medical School, University of Warwick, Coventry, United Kingdom, **3** Wolfson Institute of Population Health, Queen Mary University of London, London, United Kingdom

☯ These authors contributed equally to this work.
\* Nathan.Griffiths@warwick.ac.uk

**Data Availability Statement:** The model is available at: https://github.com/nathangriffiths/CVD-Agent-Based-Model. The underlying source data is available as described in the paper.

## Abstract

Cardiovascular disease (CVD) is the leading cause of mortality globally, and is the second main cause of mortality in the UK. Four key modifiable behaviours are known to increase CVD risk, namely: tobacco use, unhealthy diet, physical inactivity and harmful use of alcohol. Behaviours that increase the risk of CVD can spread through social networks because individuals consciously and unconsciously mimic the behaviour of others they relate to and admire. Exploiting these social influences may lead to effective and efficient public health interventions to prevent CVD. This project aimed to construct and validate an agent-based model (ABM) of how the four major behavioural risk-factors for CVD spread through social networks in a population, and examine whether the model could be used to identify targets for public health intervention and to test intervention strategies. Previous ABMs have typically focused on a single risk factor or considered very small populations. We created a city-scale ABM to model the behavioural risk-factors of individuals, their social networks (spousal, household, friendship and workplace), the spread of behaviours through these social networks, and the subsequent impact on the development of CVD. We compared the model output (predicted CVD events over a ten year period) to observed data, demonstrating that the model output is realistic. The model output is stable up to at least a population size of 1.2M agents (the maximum tested). We found that there is scope for the modelled interventions targeting the spread of these behaviours to change the number of CVD events experienced by the agents over ten years. Specifically, we modelled the impact of workplace interventions to show that the ABM could be useful for identifying targets for public health intervention. The model itself is Open Source and is available for use or extension by other researchers.

## Introduction

Cardiovascular disease (CVD) is the leading cause of mortality globally and is also the second main cause of mortality in the UK [1]. More than 6.8 million UK adults are estimated to be

**Funding:** NG, JA, and OO were funded by the Population Health Agent-based Simulation Network (£25K), https://phasenetwork.org/. The funder had no direct involvement in any aspect of the research.

**Competing interests:** The authors have declared that no competing interests exist.

living with CVD, which places a burden on individuals and their families but also places a huge demand on health services [2]. Prevention is key to reducing CVD morbidity and mortality. Four key modifiable behaviours are known to increase CVD risk (and NCD risk-more generally), namely: tobacco use, unhealthy diet, physical inactivity and harmful use of alcohol [3]. These four behaviours, and their direct metabolic sequalae (obesity, hypertension, glucose intolerance and hypercholesterolaemia), make up eight of the top ten leading risk factors for years of life lost across the UK [4].

Behaviours that increase the risk of CVD can spread through social networks because individuals consciously and unconsciously mimic the behaviour of others they relate to and admire [5, 6]. Exploiting these social influences has led to effective public health interventions such as Stoptober, in which a single collective push for smoking cessation in one month has more impact than lower-level messaging throughout the year [7]. However, interventions such as Stoptober target a single behaviour at a population level, and are informed by population level models based on techniques such as Ordinary Differential Equations (ODEs) [8] or psychological principles [9]. To enable the design of interventions that are focused towards specific groups or communities, or towards combinations of behaviours, a finer grained model of individuals, their behaviours and the spreading relationships between individuals is needed. Agent-based modelling offers a means to construct such models, and has been effectively used to model individual risk factors and for small populations [10–12].

This project aimed to construct and validate an agent-based model (ABM) of how the four major behavioural risk-factors for CVD spread through social networks in the population leading to the development of CVD, and examine whether the model could be used to identify targets for public health intervention and to test intervention strategies. Specifically, we aimed to build a city-scale ABM to model the behavioural risk-factors of individuals, their social networks (spousal, household, friendship and workplace), the spread of behaviours through these social networks, and the subsequent impact on the development of CVD. The model itself is Open Source and is available for use or extension by other researchers. The code and basic documentation for the model is available from https://github.com/nathangriffiths/CVD-Agent-Based-Model.

## Related work

### Agent based models

ABMs are a versatile modelling approach, capable of emulating numerous types of systems. In biology, ABMs have been used to model the evolution of species and how interactions with other species affect their evolution [13, 14], as well as simulating the human immune system [15, 16]. Beyond biological simulation, ABMs have commonly been used to model the spread of influence between individuals in a wide range of environments [17–19]. In these simulations there is typically a focus on the spread of information, and how it influences individual behaviour [20]. Such models have a wide range of applications, including areas such as economics, politics and epidemiology [21–23].

Measuring influence, and tracking individuals who have been influenced, is a complex process in the real-world, and to truly measure it is impossible. To compensate, a behaviour is typically used as a proxy for influence. For example, in business studies, we might track the purchasing of a particular item or, in politics, we might see which users have shared a particular news story. ABMs have also been used to model behaviours, and how they may spread through a social network [24].

When considering CVD risk-factors, the four main risk behaviours of smoking, poor diet, physical inactivity and harmful alcohol use have been shown to spread socially. Christakis and

Fowler found that obesity, a condition that is driven by behaviour, can spread in a similar fashion to a communicable disease, with the risk of becoming obese increasing when an individual has friends or a partner who is already obese [5]. Similarly, in a later study they demonstrated that while the number of smokers is decreasing, clusters of smokers are staying the same size, highlighting the impact of an individual's social network in relation to their ability to quit smoking [6]. Rosenquist *et al.* found the same was true for alcohol consumption, where an individual's social network plays a significant role [25]. In a large study in 2017, Aral and Nicolaides also demonstrated that exercise exhibits social contagion behaviour [26].

## Spreading models and networks

Due to the wide range of applications for influence modelling, there are a wide variety of spreading models, each taking into account the unique aspects of the problem it was developed to help solve. However, in the majority of cases, the model will be an adaptation of either the linear threshold model [19] or the independent cascade model [18]. It has even been demonstrated that both models are equivalent mathematically [27], although they are typically discussed separately since their abstractions attempt to represent different aspects of social influence.

In the linear threshold model [19], a node is influenced by each of its neighbours to varying degrees, as determined by the edge weights. Each node $v$ has a threshold $\theta_v$, and when the sum of the weights of the active neighbours of $v$ exceeds $\theta_v$, $v$ becomes active. The aim of this model is to capture the notion of social pressure, since as more of a node's neighbours become active, the chance of the node itself becoming active will increase.

In the independent cascade model [18], when a node $v$ becomes active it gets one chance to activate each of its inactive neighbours $w$, with some probability. Any node that becomes activated then has the same chance of passing the activation to its neighbours in the next time step. This model captures the notion of 'word of mouth' and how individuals can influence each other directly.

Both of these models are limited with respect to their focus on a single cascade (or concept) spreading through a network and that individuals, once activated (or infected), cannot become inactivated (or uninfected). To combat these limitations, models that allow for multiple concepts to spread have been proposed, often focused on epidemiology [28–31]. These models allow for multiple concepts to compete within a network, although individuals may still only be infected by a single concept. Sanz *et al.* proposed a multi-layer network model to overcome this, allowing for concepts to each spread on a single layer and making it possible for an individual to have multiple infections simultaneously [32]. These concepts could even interact and impact the ability of other concepts to spread, with an individual becoming more or less susceptible to a concept if they already had an infection with another concept.

The notion of interacting concepts has been further studied and generalised, including methods of exploiting the relationships between concepts. However, these models treat concepts in a straight forward manner, such that an individual either is infected or they are not. When we consider behaviours that contribute to CVDs, we must begin to consider the notion that concepts may have intensities, or levels, of adoption [33].

In addition to the spreading model, the topology of the network that connects agents affects how concepts spread through the population. There have been numerous studies of the properties of both real-world and synthetic networks [34], with corresponding mathematical approaches to modelling their topologies [35, 36]. Real-world networks are generally by characterised exhibiting small-world and scale-free properties [37–41]. Methods such as the use of Stochastic Block Models (SBMs) [35] and Exponential Random Graph Models (ERGMs) [36],

enable the modelling of network topologies, including small-world and scale-free synthetic networks. These methods can be instantiated to generate specific network types such as random graphs using Barabási–Albert preferential attachment [40] or small-world graphs using the Newman–Watts–Strogatz algorithm [42], both of which have been shown to mirror properties seen in real-world networks [34]. In the ABM described in this paper, we support both Barabási–Albert random graphs and Newman–Watts–Strogatz small-world graphs using the NetworkX Python package [43].

Since network topology has a clear role in influence spread, several studies have considered how to leverage features of nodes and the topology to maximise (or minimise) spread, such as identifying individuals with high degree or centrality [17, 19, 44–49]. However, these approaches assume that a small set of individuals (called seed nodes) are selected as the source for a concept spreading. Since we are modelling a population of agents, all of whom exhibit behaviours, rather than starting our simulations with a small set of seed individuals who exhibit behaviours, such approaches are not applicable. Although we note that if an intervention were to target individuals rather than behaviours, as is the case in the example intervention discussed in this paper, then seed set selection techniques would be useful. The role of topology on the spread and emergence of behaviour within a population has also been studied [50–52], affecting the speed with which emergence occurs, and showing similar results to influence maximisation research in that individuals who have high centrality or degree are more likely to influence those they are connected to [17, 44, 45].

## Model overview

Our proposed ABM consists of several key aspects, focused around a set of agents corresponding to the population being modelled. Each agent has a set of attributes, and a number of relationships with other agents, which form an underlying network of connections. It is through these connections that behaviours can spread through the population. Each agent's attributes and behaviours determine the likelihood of it suffering a cardiovascular event. The ABM begins by generating a population of agents with corresponding attributes and relationships. Note that we focus on modelling an adult population, and so the minimum age of an agent is 18. Once a population is generated the model is iterative, with each iteration representing a year, in which individuals' attributes are updated (e.g., agents get older), risk behaviours spread, and the chance of an agent developing CVD is calculated. For simplicity, if an agent develops CVD it is removed from the population. We also assume that the population is fixed and we do not model new agents joining, or agents leaving through any other means than by developing CVD, i.e., our model can be thought of as corresponding to a cohort study.

The model is highly customisable, with all the main parameters tunable by the user through the use of CSV (comma separated value) files. Thus, the user is able to specify the population size, age distribution, gender distribution, probability of marriage and the distribution of spouse age as a function of agent age, average household size, etc. In this section, we present the overall model, describing the individual attributes modelled, the construction of the social networks, the behaviour spread mechanism, and how the likelihood of developing CVD is determined. In the Experimental Methodology section below, we present a specific instantiation of the model, describing how the parameters are determined to construct a representative model of the population of an English city.

### Basic agent attributes

Each agent in the population is ascribed an age, sex and socio-economic status that is selected according to the probability distribution that is defined by the user (and supplied as a CSV to

the model). In addition, each agent has an assigned level of adoption for four behavioural risk factors, which are described in detail in 'Behaviour Spread' below.

The behavioural risk factors directly affect the chance of an agent developing CVD, modifying a base rate that is determined using an agent's sex and age. An agent's socio-economic status has a less direct impact, since it influences the agent's probability of being employed and included in a network of workplace contacts. Workplace contacts then provide additional opportunity for the agent to adopt new levels of behaviour which will then impact their chance of CVD.

While age, sex and socio-economic status are the only personal characteristics we consider in the model, the model can easily be extended to include additional attributes, such as ethnicity, etc. However, when incorporating additional attributes it is important (i) to understand and account for how they affect the development of CVD, (ii) factor in any interactions between attributes in the CSVs used to initialise the model (e.g., adding an ethnicity attribute may interact with age when defining the base likelihood of developing CVD), and (iii) the calibration method described below should be refined to account for the additional attributes.

## Agent relationships

In the real-world, an individual's behaviour with respect to the CVD behavioural risk factors can be influenced by their social contacts. For example, an individual is more likely to be a non-smoker if their household, friends, and close workplace contacts are also non-smokers. However, the strength of this influence naturally varies based on the nature of the relationship. In our ABM, we consider four types of relationship that potentially connect any two individual agents, namely *marriage, household, friendship* and *workplace*. The existence of such relationships allows one agent to exert influence upon another, with the level of influence exerted being dependent on the type of relationship shared.

For simplicity, we consider these relationships as forming a strict hierarchy, ordered according to the extent to which a given relationship allows the spread of influence, such that there can be at most one relationship between any pair of individuals. Specifically, we assume that *marriage* $\succ$ *household* $\succ$ *friendship* $\succ$ *workplace*, and so if a pair of agents could have more than one relationship they are assigned the one that is highest in the hierarchy. For example, if two individuals are friends and work colleagues, we would only model the friendship relationship. Thus, we do not treat different relationships as having a cumulative effect on behavioural influence. In the real-world, one would expect that an individual considers their spouse as a friend, but that the spousal relationship is more significant in terms of influencing the behavioural risk factors.

## Population generation

The first step in our ABM is to generate a population of agents according to the parameters supplied by the user. To create a population we begin by generating a set of households (stopping once the target population size is reached). We then create a friendship network between the agents in these households, and create workplace relationships between individuals. This process is described in more detail below. Note that all probability distributions and network parameters are supplied by the user, and in the Experimental Methodology section we describe an illustrative instantiation of these parameters.

**Marriage.** We generate the population by initially creating households, each of which is focused around a single agent. So, to begin, we generate the first agent in the first household, ascribing values for its age and sex attributes as described above. We then determine whether that agent is married, using a probability determined by their age and sex. If an agent is

married, we generate their spouse, i.e., we create a second agent in the household, again ascribing values for the age and sex attributes. The spouse's age and sex values are selected probabilistically according to the original agent's sex and age, and the probability of marriage being between individuals of the same sex. Note that for simplicity, the ABM does not distinguish between couples who are married, in civil partnership, or are common-law partners.

**Households.**   After determining whether the initial agent in a household is married, and generating their spouse if appropriate, we then select the household size according to a probability distribution. Note that since our ABM focuses on modelling adult populations, we do not consider children under 18 in the simulation, and so only adult household members are modelled. While typically less common, we allow for married couples to be in a household that contains other adults (e.g., non-dependant children over 18 years of age). We then generate the remaining agents for each household, with each new agent being ascribed a sex and age chosen according to the agent attribute probability distribution.

We consider the agents in a household to represent a fully connected network, such that each agent shares a household relationship with every other agent in the household, unless that agent is also their spouse in which case the marriage relationship is recorded since it is higher in the hierarchy. Finally, we assign all agents in the household a socio-economic status using the Index of Multiple Deprivation (IMD), probabilistically determined according to the sex and age of the initial household member. A household's IMD is a measure of the relative deprivation of the area where it is located, in terms of seven dimensions including income, employment, and education. We use the IMD measure since it is adopted and published by the Office for National Statistics for England [53]. However, the implementation of the model is structured such that it is easy to incorporate alternative measures for different geographical regions.

Once a household has been generated, we then create a new agent to act as the first agent in the next household, generating its spouse (if appropriate) and its household, repeating the process until the specified population size is reached. Since the ABM generates complete households, the overall population size may slightly exceed the target size. However, due to the scale of household sizes, this overshoot is never by a large amount and we may expect to see approximately 2–4 additional agents at most.

**Friendship.**   Once all households are generated, we have a complete population of agents with associated attributes. The next step is to generate the friendship network. We select a subset of the population to include in the friendship network, allowing for the possibility to exclude a small proportion of the agents in the population who lack social connections [54]. A friendship network is then generated in the form of a small-world network, since small-world networks have been observed to exhibit similar properties to those seen in real-world social networks [55, 56]. Rather than using a more abstract representation of networks, such as SBMs [35] or ERGMs [36] (as discussed in the Related Work section), we use specific instantiations of network generation algorithms that have been shown to produce networks with similar characteristics to real-world networks [37–41]. Specifically, the ABM allows for either a Barabási-Albert or a Newman–Watts–Strogatz generation algorithm to be used [37, 57], although the implementation can easily be changed to use other generators. Once the friendship network has been generated, we enforce the relationship hierarchy by removing any edges that connect two agents who have a marriage or household relationship.

Note that while we support the Barabási-Albert and Newman–Watts–Strogatz generation algorithms, if real-world data on friendships was available then an alternative generator could be used. In this case, a SBM or ERGM approach could be used directly, since there would be data available to calibrate the methods and parameters controlling the network generation. SBMs have previously been used to model real-world networks, particularly with respect to degree-distribution, but at relatively high computational cost [58]. It is also worth noting that

SBMs do not fully account for the scale-free aspects of real-world networks [58], and have only been applied to relatively small networks (around 10K individuals [59] where calibration data exists [60, 61], rather than up to 1.2M in our ABM).

**Workplace contacts.**    There is little literature or data relating to the topology of influence for workplace contacts. However, we aim for our ABM to support investigation into how influence from close workplace contacts might impact the spread of behaviour, and the potential nature of workplace interventions. Therefore, we include workplace relationships by generating groups of close work contacts. We take this approach for simplicity, rather than simulating agents working for a particular company along with the corresponding structures and hierarchies within those companies. Thus, the ABM is able to represent the team an individual may work within, or the close work colleagues an individual may interact with on a daily basis (but does not represent the company itself).

Each agent has a probability of being employed, determined according to their age and sex, and the IMD assigned to their household. To generate the workplace contacts we first determine each agents' employment status. From the set of employed agents we begin by selecting an agent (at random) to be the first member of a workplace contact group. The size of the agent's contact group is determined by first selecting the size of the workplace the individual belongs to, based on the distribution of workplace sizes. For simplicity we consider a workplace to be one of four sizes in terms of total employees, namely $<10$, $<50$, $<250$, or 250+. We then select the size of workplace contact group using a normal distribution with a mean determined by the size of the workplace and standard deviation determined by the range of workplace contact sizes (allowing for different workplace sizes to lead to different size groups of close contacts). This process is then repeated until all employed agents have been assigned a workplace. Note that this approach means that there may be a small number of agents (at most bounded by the workplace contact group size) whose actual workplace contact group size is smaller than that selected according to the probability distributions.

As when generating households, we assume that workplace contacts form a fully connected network, with each agent in the group being connected to every other agent. Note that again, we enforce the relationship hierarchy, and so if an agent is assigned to be a workplace contact with their spouse, a household member or friend then these relationships take precedence over the workplace relationship in terms of influence, and so the workplace relationship is ignored.

## Behaviour spread

Our ABM considers four behaviours that can be affected by social influence. These four factors, namely alcohol consumption, smoking, inactivity and poor diet are the main four factors that contribute to the risk of developing CVD [62]. In our model, each of these behaviours has three levels of potential adoption, with each subsequent level contributing more toward the likelihood of developing CVD, as discussed below. Table 1 indicates the properties that each level of each behaviour aims to represent in real-world terms.

**Table 1. The behaviour levels in the model and the real-world behaviours they represent.** Note that MVPA measures the combination of minutes of moderate and vigorous physical activity, where one minute of vigorous activity is equivalent to two minutes of moderate physical activity (i.e., 75 minutes of vigorous activity = 150 minutes of MVPA).

| Behaviour Level | Smoking | Alcohol | Diet | Inactivity |
|---|---|---|---|---|
| 0 | Never smoked | Consume $\leq$ 14 units a week | Consume $>$ 5 portions of fruit and vegetables daily | $\geq$ 150 minutes of MVPA weekly |
| 1 | Ex-smoker | Consume 15 to 49 units a week | Consume 2-5 portions of fruit and vegetables daily | 30-149 minutes of MVPA weekly |
| 2 | Current smoker | Consume $\geq$ 50 units a week | Consume $<$ 2 portions of fruit and vegetables daily | $<$ 30 minutes of MVPA weekly |

Each agent has an initial level of adoption for each behaviour, determined probabilistically according to the agent's age and sex. To simulate the effect of social influence on these behaviours, we utilise an approach based on the linear threshold influence model [63]. Each agent is given a threshold, chosen randomly using a Gaussian distribution, which represents how difficult it is for the agent to be influenced to change their level of behaviour adoption, with a higher threshold representing that an agent is harder to influence. In our instantiation of the model, discussed below, we use a mean of 0.8 and a standard deviation of 0.05 for the Gaussian distribution, as illustrative values that represent agents being somewhat resistant to influence, but these values can be changed by the user.

Each relationship type carries with it a level of influence, which is defined per relationship and behaviour level. For example, a spouse may exert an influence of strength 0.525 in regards to level 1 alcohol consumption behaviour. As discussed below, in the Experimental Methodology section, for our evaluation we generally assume a base strength of 0 for all influence exerted by workplace contacts (since there is not reliable data on the strength of this form of influence). However, we explore the impact of potential workplace interventions in which we use non-zero values for workplace influence.

The ABM progresses in discrete time steps, with each time step representing a duration of one year. In each step, we consider the incoming influence exerted on each agent by the agents they have relationships with. We sum the incoming influence from all relationships for each level of each behaviour, giving 12 values to consider (since there are 4 behaviours, each having 3 levels). For each behaviour, an agent will adopt the level of the behaviour that has exceeded the agent's threshold. If none have exceeded the threshold, the agent will maintain its previous level of adoption. If more than one level has incoming influence exceeding the threshold, then the agent's level of adoption will be chosen using a weighted probability where the probability of adopting a given level is proportional to the amount of influence being exerted.

Thus, agents can change from any level to any other level with respect to behaviours. So, for example, changing from level 0 adoption of alcohol consumption to level 2 does not require a time step spent at 1. The exception to this is level 0 of smoking adoption, which represents that an individual has never smoked. Thus, even if a smoker quits they can only become an ex-smoker, represented by level 1 of smoking adoption. Only agents that initially start with level 0 for smoking can maintain it, and once the level changes the agent can never adopt level 0 again.

## CVD risk

At the end of each time step, after we have determined which level of each behaviour each agent has adopted, we calculate the likelihood that the agent will develop CVD. This consists of two main steps, namely (i) calculating the baseline likelihood of developing CVD, and (ii) adjusting this likelihood according to the behaviours currently adopted by each agent.

The baseline CVD likelihood for an agent is calculated using an adapted version of the QRISK3 algorithm [64]. Due to the simplicity of our agents, with age and sex being the basic attributes, we simplify the QRISK3 algorithm by using the default values for BMI, cholesterol and blood pressure for males and females, as provided in the original QRISK3 algorithm. We use only the core parameters of QRISK3, and all optional modifications are ignored since our ABM does not include the corresponding factors. These could be introduced in a future extension of the model, if relevant data were identified.

Once the baseline CVD likelihood is calculated, we modify it based on the behaviour levels an agent has adopted. Each behaviour level has a risk factor (defined by the user), which the baseline CVD likelihood is multiplied by. This risk factor is also dependent on the age of the

agent, and so our ABM allows for each age range to have a unique set of risk factors. We are not able to account for short and long-term effects of a behaviour, so, for example an agent that is an ex-smoker experiences the same increase in their risk of CVD whether it was a current smoker in the previous time step, or several time steps previously. Similarly, an agent consuming 50+ units of alcohol a week experiences the same increase in their risk of CVD whether they have expressed this behaviour for one or ten time steps. There is data available that could allow this to be changed in a future extension of the model. For each agent in the population, therefore, we take their baseline CVD risk and multiply it by four factors, one for each behaviour, to give the final CVD risk for that agent. This final CVD risk factor represents the probability of the agent developing CVD in the next ten years, and so we divide it by 10 to get the probability that the agent will develop CVD in the current year. The final step is to probabilistically check each agent according to their risk to determine whether they develop CVD, and if they do, we remove them from the simulation.

## ABM simulation process

Each simulation using the ABM begins by generating a population of agents, and their connections as described above. Then, the simulation proceeds in a series of time-steps, with each iteration representing a year, in which behaviours spread through social connections, CVD risk is calculated, and any agents developing CVD are removed from the population. For each iteration of the simulation we perform the following steps.

1. Calculate the incoming influence for each agent, for each behaviour and adoption level.

2. Use the incoming influence to determine the adoption level of each behaviour for each agent.

3. For all agents, simultaneously update their behaviour adoption levels.

4. Calculate the CVD risk for each agent, using the simplified QRISK3 algorithm, modifying this risk based on an agent's age and adoption level for each behaviour.

5. Test each agent to determine whether they have developed CVD in this iteration (i.e., in this year of simulated time).

6. Agents that develop CVD are removed from the population, along with all their connections and influence.

By default, simulations in our ABM run for 10 time-steps, but this can be adjusted depending on the goals of the user running the simulation. Once complete, the ABM outputs a number of metrics, including the prevalence of level 2 behaviours in the remaining population and the number of CVD events per 1, 000 person years for each age range and sex.

## Limitations of the ABM model

Currently, beyond the data used for the parameters, the ABM simulation process itself has three main limitations that may affect its utility in some scenarios.

First, our ABM is a closed system, meaning that no new agents are introduced once a simulation begins. Furthermore, we do not update relationships except in the case of an agent being removed. Therefore, the population is fairly static, which may impact how the behaviours spread and, in turn, impact the rate of CVD. Introducing new agents could most naturally be done by generating new agents of the age of 18 (i.e., the youngest age considered by our model), but placing such agents into households, friendships and workplaces is complex and there is insufficient real-world data to inform how to do this in a realistic manner. Similarly,

there is a lack of real-world data on the nature rate of relationships forming and breaking in a population (according to sex, age, IMD, etc.), and therefore we did not include this in the model. Therefore, the population in the ABM can be considered as a cohort that persists throughout the simulation process.

Incorporating a dynamic friendship network or dynamic workplaces would require data on how connections are removed and added within the population [34], but such data is not available at a population level. While there has been previous work on modelling the change in friendship networks over time such approaches either consider abstract networks [37, 38] or tend to be small scale ($<$ 1000 individuals) and either require the full friendship network as input [65] or to be based on social media data [66, 67]. We do not have access to a city-scale real-world friendship network, and we do not use social-media data due to access issues and, moreover, the skewed nature of the types of friendships this would record. Similarly, we do not have real-world data to support modelling agents changing workplace contact groups. Therefore, we use static networks in our ABM (other than removal after a CVD event) as is common in other work modelling influence spread [17, 19, 32]. As discussed in the Experimental Methodology section, we use a Newman-Watts-Strogatz small-world model for friendship since this has been shown to be comparable with real-world friendship networks [34, 65–67]. However, since the friendship network in the model is built using NetworkX [43], if such real-world data were to become available this could easily be incorporated using the read-from-file functions in NetworkX.

Second, the model does not include agents under the age of 18 (although one might imagine that children would be present in some of the households) or over the age of 90. Our decision to exclude modelling individuals age under 18 was due to a lack of real-world data, particularly on how to allocate such individuals to households, friendship groups, and workplaces when they become adults. In reality, children are likely to have an impact on a parent's behaviours and the nature of their social networks, which in turn will impact on the likelihood of a parent developing CVD. However, our ABM does not model the influence or impact of children on their parents. We excluded individuals over 90 because of their small numbers, the likelihood that they do not live in private households, the probability that their behaviour is less likely to change, and their risk of CVD is less likely to be altered by any behaviour change.

Finally, our ABM assumes that any CVD event is fatal and removes the agent from the population. This aligns with the data presented by Hippisley-Cox *et al.*, where they focus on fatal CVD events [64]. Similarly, we assume that a CVD event only affects the agent concerned. Clearly this does not capture the full range of real-world CVD events and their implications. For example, a non-fatal CVD event could prompt a large behavioural shift in an individual, and perhaps also their social contacts, which could then spread. Our model does not capture such impacts, although it would be easy to modify the implementation to include this effect, at least in a rudimentary way, if suitable real-world data was identified on the social effects of an agent experiencing a non-fatal or fatal CVD event.

## Experimental methodology

Our proposed ABM and simulation process, described in the previous section, is generic, and can be used to model any population in terms of city size (noting that we have only tested populations $\leq$ 1.2M agents), individual and household characteristics, network connections etc. There are two main aspects to the simulation that can control the model, namely (i) the set of probability distributions that describe the population, and (ii) the extent to which the different relationships influence each behaviour. To facilitate easy configuration of these modifications, the implementation of the simulator uses a number of CSV files to control the various

experimental parameters. For the experimental results discussed in this paper, we based our probability distributions and demographics on the English (where possible) or UK population, utilising national census and government data where possible, and we configured the influence strength for the different relationships using empirical calibration, as discussed below. The simulation is released under the GNU GPL licence and is available at: https://github.com/nathangriffiths/CVD-Agent-Based-Model.

## Simulation parameters

To ensure that the modelled population is representative of the general population of England, we followed the principle of using the most up-to-date English data where possible. We utilise census data for England in order to determine the probability distributions for both the sex and age of the agent [68], with socio-economic status being based on the index of multiple deprivation [53]. The population data for England and Wales also allows us to determine the probability that an agent is married [69], and we base our ratio of heterosexual to same-sex marriage on data for England and Wales [69]. To determine the probability distribution of a potential spouse's age we processed the census data [70] to extract the required probabilities.

We base our household size distribution on the average size of a UK household (data were not available for England), including allowing for married couples to have housemates [71]. The remaining relationships are more difficult to establish, as there is no nationally representative data available. To generate the friendship network, the simulation uses the Newman–Watts–Strogatz model [42], with a mean degree of 6, and with 7% of the population excluded (i.e., representing that 7% of the population do not have close friends) to approximate real-world characteristics [54].

For work contacts there is little real-world data, and so for simplicity we include two basic scenarios that assume every workplace has the same number of average contacts, either 4 or 21 [72, 73]. When we construct workplace contact groups in the simulation, we select a group size using a normal distribution with either 4 or 21 as the mean size for that simulation. The workplace contact network is included to support our investigation into a potential intervention, discussed in the Example Intervention section below, and so, given the lack of real-world data and this aim, we do not include workplace contact influence in calibrating the model.

To determine the prevalence of the different behaviour levels, we mainly utilised the most recent Health Survey for England data published at the time, though for the levels of physical activity the Sport England Data was used [74–76]. Note that this may not be representative of usual physical activity levels due to potential confounding effects of COVID lockdowns and behavioural changes.

Each risk behaviour modifies an individual's chance of developing CVD, depending on the level of that behaviour, the age of the agent and the agent's sex. We base the value of this modification on data as reported in [77–81].

## Calibration of influence strength

In this paper, we have presented an ABM which can be used to model a city-scale population, in terms of CVD events. Since the four behaviours we consider, which can spread through the social networks in the model, impact the likelihood of a CVD event, the behaviour spread should result in the modelled rate of CVD events matching observed data, such as that reported by Hippisley-Cox *et al.* [64]. Thus, we needed to calibrate the influence strength associated with each behaviour. In this section, we give an overview of the calibration process used in our instantiation of the model, which generated the results we discuss below. The calibration process can be applied to different configurations of the model, by (i) modifying the initial

values to use different base data on social relationships, and (ii) changing the observed data source used to calibrate against.

Initially, we based the influence strength exerted by each relationship on results reported in several studies of the impact of social relationships on behaviours [6, 25, 82, 83]. These studies relate to one or more of the behaviours considered in our ABM, and how social relationships impact their adoption. There is a focus in these studies on the effect of partners/spouses and close contacts, which in some studies is defined as corresponding to friends while in other studies is defined as siblings or other family. Translating the data from these studies into influence strengths is not exact and requires subjective interpretation, and our initial experiments found the incident rate in each age range to be significantly incorrect. Therefore, to align the ABM output with the Hippisley-Cox *et al.* [64] data, we needed to explore the state space.

We initially ignored work contacts and treated their influence as 0 for all behaviours. The role of work contacts is discussed further in the Example Intervention section below. This leaves three relationships (spouse, household, friendship) which exert influence on the four risk factor behaviours, each of which has three potential levels of adoption in the ABM. This gives 36 continuous variables to explore. To explore this complex space, we utilised a random-restart hill-climbing approach [84]. We defined an initial range for our variables, set each one to a random value within that range and performed five simulations. We compared the incident rate per age group in each simulation to the Hippisley-Cox *et al.* data, summing the absolute difference of each comparison to give a value representing how close the simulation was to the observed data. We used the average of this value over five simulations to represent the fitness of the current variable selection, with the goal being to get the absolute difference to approach 0. We then adjusted the values slightly, adding or subtracting a random value between 0 and 0.05, and tested this new parameter set. If it improved upon the previous model, we used these new values as the baseline for our adjustment. Otherwise, we modified the original values again.

After we tested a given set of parameters, to ensure we explored a reasonable proportion of the state space, there is a small probability of selecting a new random set of values, instead of adjusting the existing set of parameters. This probability was increased each time we did not improve upon our current best set of values, as this could indicate we were stuck in a local minima. After a given number of iterations, we output the best performing set of variables found so far.

In parallel, we performed 500 searches, each one hill-climbing for 100 iterations. At the end of this, we compared the 500 sets of parameters and selected the best performing set. We then further refined our search by using this parameter set as the midpoint for our range of values in a new round of hill-climbing. The initial values were randomly chosen to be any value within 0.1 of the best performing parameter set from the previous set of searches. This process was repeated 5 times, giving a total of 250, 000 parameter sets tested. Although further refinement is possible, we adopted the parameters found after this process. We discuss the performance of our chosen parameter set in the Baseline Model section below. After this calibration process, the final influence strengths exerted by spouses, housemates and friends for each level of each behaviour can be seen in Table 2.

**Table 2. Strength of influence exerted by each relationship for each behaviour level, after calibration.**

| Relationship | Smoking | | | Alcohol | | | Diet | | | Inactivity | | |
|---|---|---|---|---|---|---|---|---|---|---|---|---|
| Level | 0 | 1 | 2 | 0 | 1 | 2 | 0 | 1 | 2 | 0 | 1 | 2 |
| Marriage | 0.554 | 0.691 | 0.635 | 0.316 | 0.525 | 0.755 | 0.331 | 0.659 | 0.773 | 0.472 | 0.959 | 0.255 |
| Household | 0.193 | 0.099 | 0.497 | 0.252 | 0.46 | 0.785 | 0.266 | 0.192 | 0.522 | 0.037 | 0.589 | 0.373 |
| Friendship | 0.276 | 0.104 | 0.359 | 0.229 | 0.371 | 0.749 | 0.169 | 0.53 | 0.594 | 0.124 | 0.487 | 0.496 |

## Experimental parameters

For the simulation results reported in the following section, we use a baseline population of 350K agents, unless stated otherwise, in order to simulate a population that is similar in size to the city of Coventry [85]. Furthermore, we used the probability distributions and influence levels discussed above, with the goal of representing the population of England. Each simulation configuration was run for 10 time steps, representing 10 years.

Due to the stochastic nature of the model, we naturally must perform a simulation with the same parameters multiple times. In order to gain an idea of the average performance, all experiments discussed involved 100 unseeded simulations of each parameter combination. We include the standard deviation in all relevant tables to demonstrate variance that could be expected from the model, allowing us to demonstrate the consistency of the model.

For the experiments we present below, in the Results section, we adjust these parameters to fully explore the capabilities of the ABM. The exception to this is the first experiment (described in the Baseline Model section), where we evaluate the ABM performance against the real-world data observed by Hippisley-Cox *et al.* [64].

To examine the model's expressiveness, we modify the influence strength of different relationships (discussed in the Varying Influence section). We maximise the level 0 and level 2 influence strengths alternately for each behaviour to examine the range of CVD events that the model can represent, and so explore the model sensitivity.

To examine the scalability of the model, and ensure that CVD incident rates remain constant as the population size increases, we varied the population size in steps of 100K. Our experiments considered populations ranging between 100K and 1.2M (this upper bound was chosen as it represents the size of the largest local authority area, and the second largest city in England, Birmingham, West Midlands). The results of this experiment are discussed in the Population Size section.

Finally, we explore the possible impact of targeted interventions by leveraging the work contact relationship. In the baseline model, work contacts do not exert any influence over behaviours. However, in our example intervention we assume that they exert influence over the chosen behaviour levels. We set the work contact influence strength to be 50% of the strength of the friendship relationship, based on the assumption that work contacts are less influential than close friends. Recall that due to the hierarchical nature of the model, any friends who are also work contacts will already exert influence according to the friendship relationship. We chose three interventions, focusing on diet, inactivity and both simultaneously. In all cases, work contacts exert influence in level 0 and level 1 of the intervention behaviours. The results for this intervention exploration are discussed in the Example Intervention section.

## Results

### Baseline model

To evaluate the performance of the baseline model, we generated a population of 350K agents, using probability distributions based on UK population data and utilising the influence strengths for each relationship as described above in the Calibration of Influence Strength section. We selected the population size of 350K since it is representative of the English city of Coventry, our local city and a city with numerous health concerns, and a life expectancy 9.5 years lower than the national average [86]. As such, it represents a population that may benefit greatly from effective interventions. For this initial evaluation, we ran 100 simulations for 10 time steps. Each simulation had a new population generated, allowing us to understand the expected average performance of the model.

**Table 3. Summary of model results, showing number of CVD incident cases, person years and rate per 1,000 person years, with standard deviations in parentheses.** Results are averaged over 100 runs for a population of 350,000 agents, using the parameters described in the Experimental Methodology section.

| Age group (years) | Women | | | Men | | |
|---|---|---|---|---|---|---|
| | Incident cases | Person years | Rate | Incident cases | Person years | Rate |
| 25-29 | 42.5 (6.3) | 121150.8 (693.5) | 0.4 (0.1) | 57.8 (7.8) | 129195.9 (610.8) | 0.4 (0.1) |
| 30-34 | 99.2 (10.5) | 140470.4 (827.4) | 0.7 (0.1) | 179.0 (13.6) | 146932.3 (731.7) | 1.2 (0.1) |
| 35-39 | 112.3 (11.4) | 146838.1 (710.8) | 0.8 (0.1) | 220.8 (15.7) | 148061.9 (767.6) | 1.5 (0.1) |
| 40-44 | 186.6 (13.9) | 145552.4 (678.5) | 1.3 (0.1) | 388.6 (20.4) | 142941.3 (745.5) | 2.7 (0.1) |
| 45-49 | 292.1 (17.6) | 140962.6 (685.4) | 2.1 (0.1) | 616.2 (25.9) | 137299.0 (699.6) | 4.5 (0.2) |
| 50-54 | 476.4 (20.3) | 146341.0 (737.6) | 3.3 (0.1) | 965.9 (31.4) | 140708.9 (795.2) | 6.9 (0.2) |
| 55-59 | 784.1 (31.1) | 154644.8 (816.9) | 5.1 (0.2) | 1466.8 (38.6) | 146244.4 (878.5) | 10.0 (0.3) |
| 60-64 | 1130.8 (33.8) | 146403.8 (806.0) | 7.7 (0.2) | 1959.8 (51.1) | 136441.5 (802.4) | 14.4 (0.4) |
| 65-69 | 2069.1 (50.3) | 125789.7 (680.7) | 16.4 (0.4) | 3266.9 (69.5) | 114552.6 (594.1) | 28.5 (0.6) |
| 70-74 | 2612.7 (52.2) | 110409.2 (728.1) | 23.7 (0.5) | 3651.6 (79.6) | 97337.2 (565.8) | 37.5 (0.8) |
| 75-79 | 3301.8 (67.6) | 96675.4 (661.7) | 34.2 (0.8) | 4065.8 (77.3) | 81304.4 (596.8) | 50.0 (0.9) |
| 80-84 | 3477.4 (60.2) | 71985.1 (569.1) | 48.3 (1.0) | 3713.8 (72.0) | 56142.6 (452.8) | 66.2 (1.3) |
| total | 14584.9 (159.5) | 1547223.2 (2603.3) | 9.4 (0.1) | 20553.1 (218.6) | 1477162.1 (2703.9) | 13.9 (0.2) |

Table 3 shows the baseline output of the model. This shows the expected epidemiological patterns for CVD risk, i.e.: higher rates for older age-groups and for men compared with women, which are well-known trends. Table 4 shows how the rate compares to the observed data as reported by Hippisley-Cox *et al.* [64]. The difference between the observed data and modelled rate is shown in Fig 1. These results show that the ABM is able to model a population of size 350K, producing output that aligns reasonably with the observed data. The average difference from the observed data is 0.68 CVD events per 1,000 person-years for Women and −0.48 CVD events per 1,000 person-years for Men, with a maximum difference of 3.64 CVD events per 1,000 person-years for Women and −7.86 CVD events per 1,000 person-years for Men. The modelled data is slightly further from the observed data for the middle age-groups (ages 55-64), although still acceptable, which may suggest there are specific risk-factors in these age-groups that we do not currently model, or model quite as well.

**Table 4. Comparison of the rate per 1,000 person years as generated by the model output and the observed data as reported by Hippisley-Cox *et al.* [64].** The modelled rate is averaged over 100 runs for a population of 350,000 agents, using the parameters described in the Experimental Methodology section.

| Age group (years) | Hippisley-Cox et al. Observed Rate | | Modelled Rate | | Difference | |
|---|---|---|---|---|---|---|
| | Women | Men | Women | Men | Women | Men |
| 25-29 | 0.24 | 0.40 | 0.35 | 0.45 | -0.11 | -0.05 |
| 30-34 | 0.49 | 0.99 | 0.71 | 1.22 | -0.22 | -0.23 |
| 35-39 | 1.02 | 2.12 | 0.76 | 1.49 | 0.26 | 0.63 |
| 40-44 | 1.90 | 3.99 | 1.28 | 2.72 | 0.62 | 1.27 |
| 45-49 | 3.20 | 6.65 | 2.07 | 4.49 | 1.13 | 2.16 |
| 50-54 | 4.83 | 9.86 | 3.26 | 6.86 | 1.57 | 3.00 |
| 55-59 | 7.47 | 14.18 | 5.07 | 10.03 | 2.40 | 4.15 |
| 60-64 | 11.36 | 19.69 | 7.72 | 14.36 | 3.64 | 5.33 |
| 65-69 | 17.13 | 26.55 | 16.45 | 28.52 | 0.68 | -1.97 |
| 70-74 | 25.08 | 35.48 | 23.66 | 37.52 | 1.42 | -2.04 |
| 75-79 | 35.11 | 45.16 | 34.15 | 50.01 | 0.96 | -4.85 |
| 80-84 | 48.02 | 58.29 | 48.31 | 66.15 | -0.29 | -7.86 |
| total | 6.19 | 8.18 | 9.43 | 13.91 | -3.24 | -5.73 |

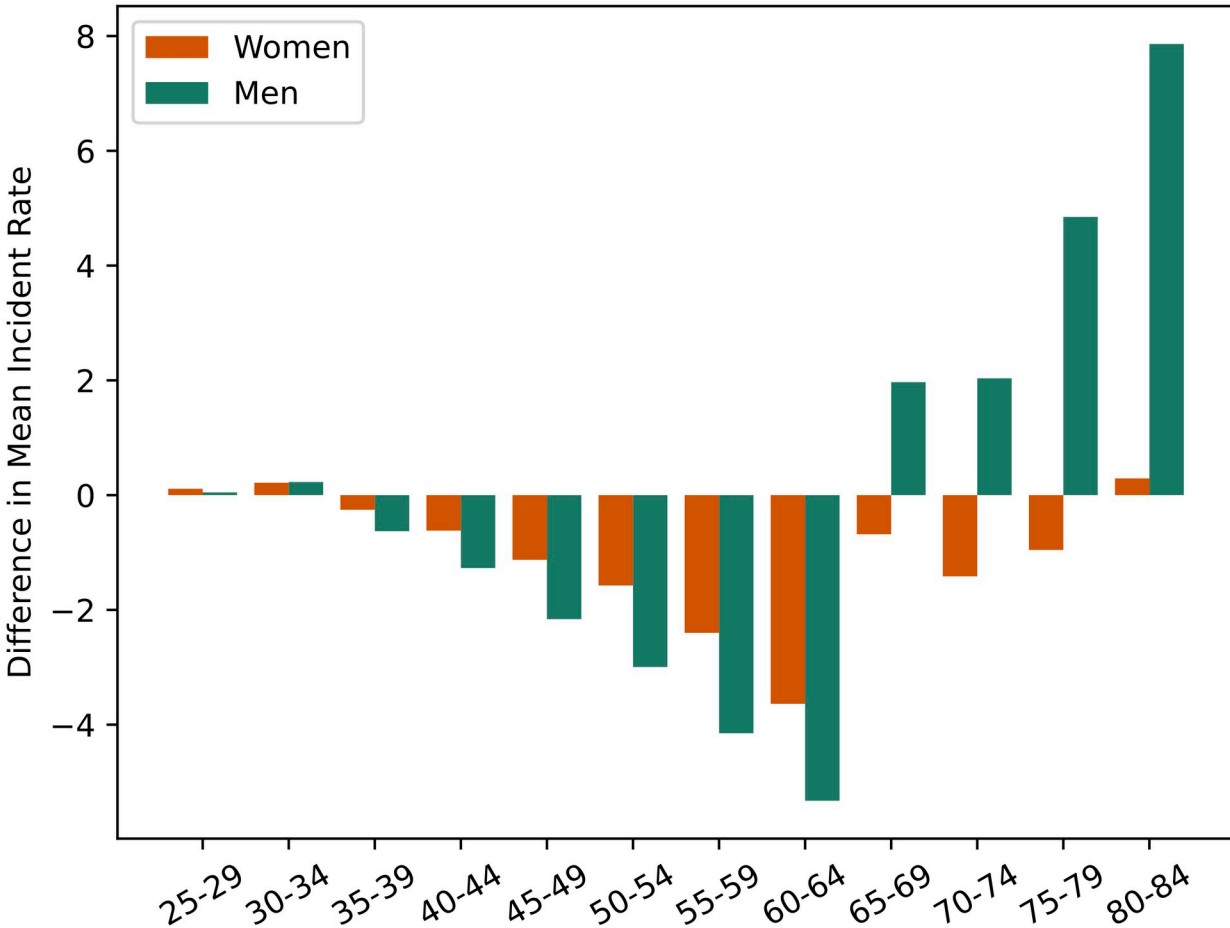

**Fig 1. The difference between observed data as reported by Hippisley-Cox *et al*. [64] and the modelled rate.** The difference between observed data as reported by Hippisley-Cox *et al*. [64] and the modelled rate (averaged over 100 runs for a population of 350,000 agents, using the parameters described in the Experimental Methodology section).

## Varying influence: Model sensitivity

To evaluate the sensitivity of the model, we utilise a number of extreme cases. This will allow us to evaluate the impact individual factors have on our model, and help to determine the range of scenarios we may be able to simulate through such processes as interventions and protective behaviours.

In a given scenario, we maximise the influence exerted by each relationship for Level 0 (minimising the risk factor) or Level 2 (maximising the risk factor) of a chosen risk factor. Then, as a final scenario, we maximise or minimise all risk factors simultaneously. All other influence strengths remain as set in our initial influence calibration. For each of these 10 scenarios, we ran 100 simulations for 10 time steps on a population size of 350K. Again, for each simulation we generated a new population.

Table 5 shows the impact of maximising the influence for either Level 0 (i.e., minimum risk) or Level 2 (i.e., maximum risk) for each of the risk factors and all factors combined on the rate per 1,000 person years. We can observe that the model has different levels of sensitivity to the different behaviours, with the least change seen in varying alcohol consumption where we see ≈ 1 per cent change in our model output (CVD events over 10 years). This jumps

**Table 5. The effect of influence for Level 0 (i.e., minimum risk) and Level 2 (i.e., maximum risk) for each of the risk factors and all factors combined on the rate per 1,000 person years.** Results are averaged over 100 runs for a population of 350,000 agents, using the parameters described in the Experimental Methodology section. The standard deviation is in parentheses.

| Max. influence | Alcohol consumption | | Healthy diet | | Exercise activity | | Smoking | | All factors | |
|---|---|---|---|---|---|---|---|---|---|---|
| | Women | Men | Women | Men | Women | Men | Women | Men | Women | Men |
| Level 0 | 9.0 | 13.4 | 8.2 | 12.2 | 8.0 | 11.9 | 5.8 | 9.1 | 4.0 | 6.3 |
| | (0.1) | (0.15) | (0.13) | (0.16) | (0.12) | (0.16) | (0.06) | (0.07) | (0.05) | (0.06) |
| Level 2 | 10.1 | 14.9 | 9.6 | 14.2 | 9.6 | 14.1 | 19.0 | 26.2 | 21.3 | 29.2 |
| | (0.12) | (0.17) | (0.12) | (0.16) | (0.13) | (0.16) | (0.12) | (0.13) | (0.13) | (0.13) |

dramatically when looking at smoking, as we can see a ≈ 15 per cent change in CVD events over 10 years for both women and men.

Furthermore, this sensitivity analysis suggests that, when considering possible factors to target for a behavioural intervention, we may wish to target smoking in order to see the most benefit to the population.

## Population size: Model scalability

To evaluate whether the model accuracy remained stable as the population size increased, we ran a number of simulations for population sizes between 100K and 1.2M, increasing in steps of 100K. We also plot the results for a population of 350K, which corresponds to the baseline model discussed above, and is the population size used in our other experiments.

The simulations used the probability distributions and parameters described in the Simulation Parameters section, along with the default influence strengths for each relationship. For each population size, we performed 100 simulations with 10 time steps, with each simulation run having a new population generated.

Fig 2 shows how the number of CVD incidents rises as the population size is increased. The number of incidents increases linearly with the number of agents, which indicates that the model is performing as expected. Note that the standard deviation is too small to be visible on the plot and reduces as the population size increases, starting at 1.09% and 1.1% of incidents at 350K agents reducing to 0.07% and 0.58% at 1.2M agents for women and men respectively. The incident rate is constant for all population sizes at 9.4 for women and 13.9 for men (with a standard deviation of 0.1, except for men in populations of 350K and 400K where the standard deviation is 0.2).

## Example intervention: Workplace diet and activity

As discussed above, in the previous experiments we have made the assumption that work contacts do not exert any influence toward any of the risk factors. This allows us to introduce influence in a workplace as a means of evaluating the effectiveness of workplace interventions within the model. For this evaluation, we settled on three different interventions; one targeting diet, one targeting inactivity and one targeting both. In a given intervention, we allow work contacts to exert influence related to Level 0 and Level 1 for the relevant risk factors. For our purposes, we set influence strength for the work contact relationship to be half of the strength of the friendship relationship for the corresponding behaviours. All other influence strengths for the workplace contact relationship remain at 0. Furthermore, we also adjust the rate of adoption for the intervention, which defines the percentage of workplaces that will be involved in the intervention. We evaluate our interventions with adoption rates of 0% to 100% in steps of 25%.

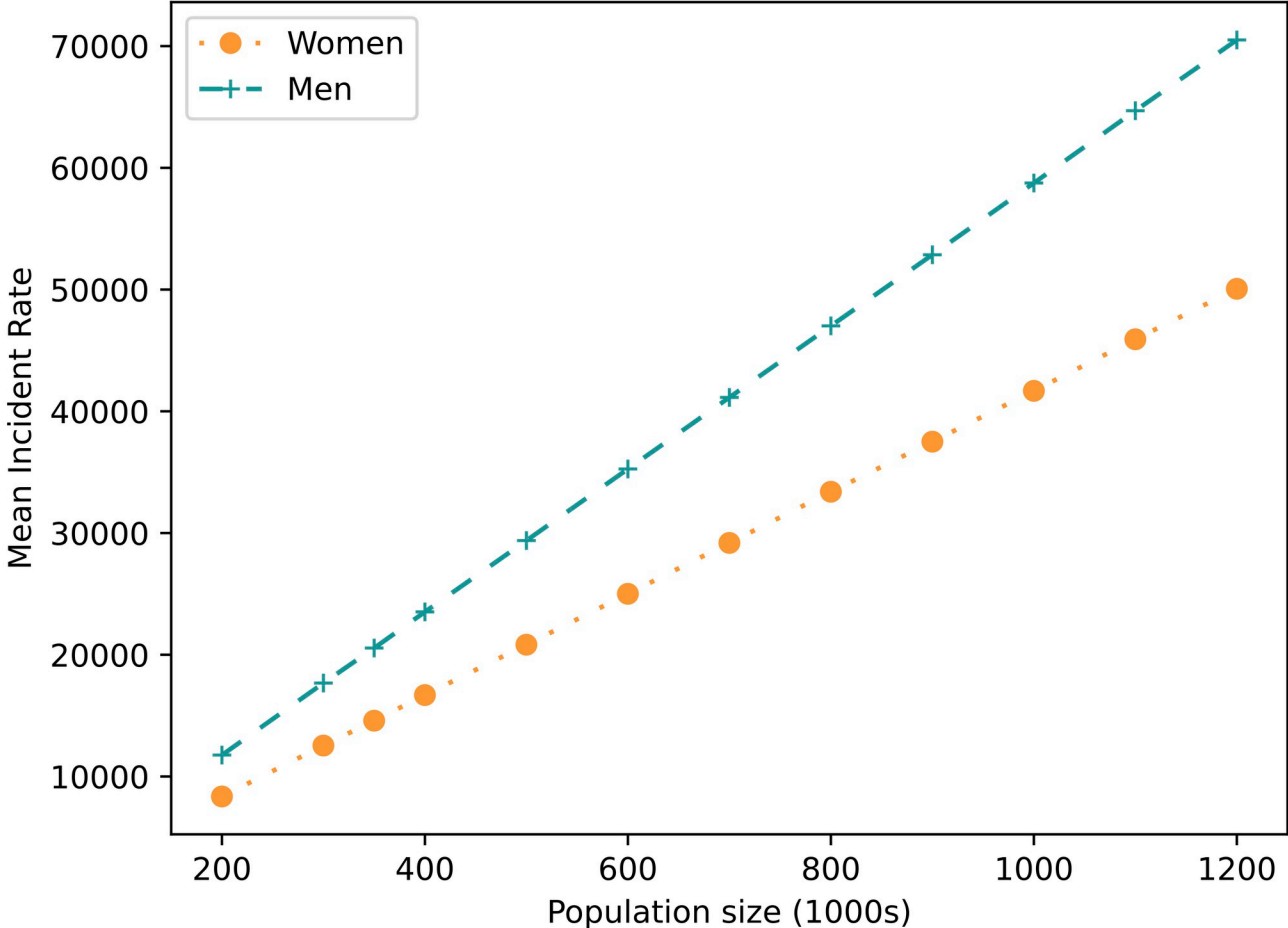

**Fig 2. Mean incident rate as population size increases.** Mean incident rate as population size increases. Results are averaged over 100 runs using the parameters described in the Experimental Methodology section (with no interventions).

All simulations used 10 time steps and a population size of 350K, with a new population generated for each simulation. For each of the interventions, we also ran them for an average number of workplace contacts of 4 and 21, as discussed in our Simulation Parameters section. Finally, for each combination of parameters (adoption rate, intervention type and average workplace contacts), we ran 100 simulations.

Table 6 shows the impact of workplace interventions to promote improved diet and physical activity levels, for workplace contact sizes of 4 and 21. The lowest rate for each intervention is highlighted in bold. As is expected, adoption of the intervention by 100% of workplaces has the largest impact in all cases (with the exception of 75% adoption for a diet intervention with a mean size of 4 workplace close contacts). It can also be seen that promoting activity has a greater effect that diet alone, and that the combination of influencing diet and inactivity has the largest effect.

## Conclusion

We have constructed an agent-based model of how the four major behaviour risk-factors for CVD spread through social networks, leading to the development of CVD at a rate that is

**Table 6. The effect of workplace interventions to promote improved diet and activity levels, for workplace contact sizes of 4 and 21, indicated in parentheses.** Results are averaged over 100 runs for a population of 350,000 agents, using the parameters described in the Experimental Methodology section, with the standard deviation in parentheses. The lowest rate for each intervention is highlighted in bold.

| Intervention | Adoption | | | | | | | | | |
|---|---|---|---|---|---|---|---|---|---|---|
| | 0% | | 25% | | 50% | | 75% | | 100% | |
| | W | M | W | M | W | M | W | M | W | M |
| Diet (4) | 9.4 | 13.9 | 9.3 | 13.8 | 9.4 | 13.8 | **9.3** | **13.7** | **9.3** | **13.7** |
| | (0.1) | (0.2) | (0.11) | (0.16) | (0.13) | (0.17) | **(0.12)** | **(0.14)** | **(0.11)** | **(0.15)** |
| Diet (21) | 9.4 | 13.9 | 9.2 | 13.6 | 9.2 | 13.6 | 9.2 | 13.5 | **9.1** | **13.5** |
| | (0.1) | (0.2) | (0.13) | (0.15) | (0.13) | (0.17) | (0.13) | (0.16) | **(0.12)** | **(0.14)** |
| Inactivity (4) | 9.4 | 13.9 | 9.4 | 13.8 | 9.3 | 13.8 | 9.3 | 13.7 | **9.2** | **13.6** |
| | (0.1) | (0.2) | (0.13) | (0.16) | (0.13) | (0.16) | (0.14) | (0.17) | **(0.12)** | **(0.17)** |
| Inactivity (21) | 9.4 | 13.9 | 9.0 | 13.3 | 9.0 | 13.3 | 8.9 | 13.2 | **8.9** | **13.1** |
| | (0.1) | (0.2) | (0.12) | (0.15) | (0.11) | (0.15) | (0.12) | (0.14) | **(0.11)** | **(0.15)** |
| Both (4) | 9.4 | 13.9 | 9.2 | 13.6 | 9.2 | 13.6 | 9.1 | 13.5 | **9.0** | **13.4** |
| | (0.1) | (0.2) | (0.12) | (0.16) | (0.1) | (0.17) | (0.13) | (0.17) | **(0.13)** | **(0.15)** |
| Both (21) | 9.4 | 13.9 | 8.7 | 12.9 | 8.7 | 12.9 | 8.6 | 12.8 | **8.6** | **12.7** |
| | (0.1) | (0.2) | (0.12) | (0.15) | (0.12) | (0.16) | (0.11) | (0.16) | **(0.13)** | **(0.17)** |

realistic compared with observed data and which scales up to a population size of 1.2M individuals, equivalent to the largest local authority area in England. We have demonstrated that there is scope for modelled interventions targeting the spread of these behaviours to change the number of CVD events experienced by the agents over 10 years, and specifically that modelling the impact of workplace interventions suggests that this model could be used to identify targets for public health intervention. There is scope for us, or others to extend this model in the future. For example, agents could be given new attributes, such as ethnicity, to enable exploration of inequalities by any new attribute. Agents could express other risk-factors including behaviours, or clinical risk-factors for CVD such as mental illness, lupus or rheumatoid arthritis. Agents could develop further health outcomes; as well as developing CVD they could also develop diabetes, cancer and/or chronic obstructive pulmonary disease. In addition, future work could allow us to make recommendations to public health practitioners and policy-makers about efficient targets for intervention.

## Author Contributions

**Conceptualization:** James Archbold, Nathan Griffiths, Oyinlola Oyebode.

**Data curation:** Sophie Clohessy, Deshani Herath, Oyinlola Oyebode.

**Funding acquisition:** James Archbold, Nathan Griffiths, Oyinlola Oyebode.

**Investigation:** James Archbold, Nathan Griffiths.

**Software:** James Archbold, Nathan Griffiths.

**Writing – original draft:** James Archbold, Nathan Griffiths, Oyinlola Oyebode.

**Writing – review & editing:** James Archbold, Nathan Griffiths, Oyinlola Oyebode.

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
