## [Decision Letter · Decision Letter 0]

29 Jan 2024

PONE-D-23-34732An Agent-Based Model of the Spread of Behavioural Risk-Factors for Cardiovascular Disease in City-Scale PopulationsPLOS ONE

Dear Dr. Griffiths,

Thank you for submitting your manuscript to PLOS ONE. After careful consideration, we feel that it has merit but does not fully meet PLOS ONE’s publication criteria as it currently stands. Therefore, we invite you to submit a revised version of the manuscript that addresses the points raised during the review process.

We look forward to receiving your revised manuscript.

Kind regards,

Keumseok Peter Koh

Academic Editor

PLOS ONE

Journal Requirements:

Reviewers' comments:

Reviewer's Responses to Questions

**Comments to the Author**

1. Is the manuscript technically sound, and do the data support the conclusions?

Reviewer #1: Yes

Reviewer #2: Yes

2. Has the statistical analysis been performed appropriately and rigorously? 

Reviewer #1: Yes

Reviewer #2: Yes

3. Have the authors made all data underlying the findings in their manuscript fully available?

Reviewer #1: Yes

Reviewer #2: Yes

4. Is the manuscript presented in an intelligible fashion and written in standard English?

Reviewer #1: Yes

Reviewer #2: Yes

5. Review Comments to the Author

Reviewer #1: Thank you for the opportunity to review the article “An Agent-Based Model of the Spread of Behavioural Risk-Factors for Cardiovascular Disease in City-Scale Populations. It is a well-written paper strongly founded on robust predictive modelling. I greatly appreciate the efforts of the authors. However, I have a few suggestions to strengthen the article further.

I have a few suggestions and observations related to the model

1. I see a possibility of extending the simulation with more factors, like

a. Common mental disorders like stress, anxiety, and depression have shown a significant association with behavioural risk factors. Therefore, how does the model address these intervening factors? It is not clear from the manuscript.

b. The genetic loading of CVD is not included in the model, as it could be a confounder. To my knowledge, behavioral aspects are only one of the various genetic, cognitive and social risk factors of CVD

2. Comparing with a network model like the Stochastic Block Model (SBM)/ or a centrality measure that considers influence strength would enhance the model's robustness by capturing the higher likelihood of interactions within the same neighbourhood.

3. The manuscript does not account for temporal dynamics, such as the short-term and long-term effects of behaviours on cardiovascular health.

4. Conduct sensitivity analyses to assess the impact of varying parameters on model outcomes

Reviewer #2: This is a clear and well written text that describes an agent based model of the reinforcement or mitigation of CVD behavioral risk factors by spousal, household, friendship and workplace networks. The model is well described and compelling with one drawback that is leading me to recommend major revision. The various networks are created during initialization and then remain static over the course of the simulated 10 year run. The static nature of the networks seems perhaps reasonable for spouse and household, but less so for friendship and workplace networks. The authors address this shortcoming in the "Limitations of the ABM Model" section, but not satisfactorily. The strength and central focus of the paper is the network mediated influence and stating that "the population in the ABM can be considered as a cohort that persists throughout the simulation process" undermines that focus. If the CVD behavioral risk factors are network mediated then the static nature cannot be dismissed so easliy. More needs to be said about why the static nature of the networks doesn't break the model, perhaps with additional experiments that do change the networks to illustrate what sort of effect, in the abstract, this might have. In addition, workplace assignment as described in the synthetic population literature may also offer some insight into workplace turnover. With respect to dynamic networks, the authors may also want to look at "ERGMs" (https://eehh-stanford.github.io/SNA-workshop/ergm-intro.html). More specific comments follow.

In the Related Work section, some brief discussion of synthetic population construction would be useful.

line 38. "the evolution", maybe better as "their evolution" or "the evoluion of ...".

l. 65 The two influence models need citations. They are cited later, but this is their first mention, I think.

In the "Basic Agent Attributes" section, it would be useful to explicitly mention that the agent's have the risk factor attributes as well as age, sex, and socio-economic status. This is obvious once the model is further described, but would be good here as well to avoid any confusion. With respect to the attributes, the authors state "these attributes have a significant effect on an agent’s likelihood of developing CVD, and are considered in combination with the impact of the behavioural risk factors described below". Further in the text, when creating the population and assigning CVD risk factors, age and sex are used, but it's unclear how socio-economic status (IMD) effects CVD in the model.

With respect to behavior spread, why 1 year time steps?

l. 371 - no need to unpack CSV, already seen on page 12.

l. 408 - Missing dash in Hippisley Cox. This occurs in other places as well.

Calibration Section - some text about the stochastic nature of the model, and the range of variation between runs with different seeds, and how many runs are necessary to characterize the model's output would be useful, assuming this applies.

Results - some text about why some age ranges do better / worse that others would help to explain the results.

Intervention - why does the baseline model against which intervention is applied have no workplace influence. More accurate / realistic would be a calibrated real-world reasonable level of workplace influence that is then modified by the intervention, unless the experiment is only to show that an intervention can have an effect in the abstract. If it's the latter, then that should be made more explicit.

l. 507 -- isn't the maximum absolute difference 7.86?

Is citation 39, the same as 3?

6. PLOS authors have the option to publish the peer review history of their article (what does this mean?). If published, this will include your full peer review and any attached files.

Reviewer #1: No

Reviewer #2: No

---

## [Author Response · Author response to Decision Letter 0]

27 Feb 2024

Please see the "Response to reviewers" file for a detailed response. We believe that we have addressed the Reviewers' concerns.

---

## [Decision Letter · Decision Letter 1]

19 Apr 2024

An Agent-Based Model of the Spread of Behavioural Risk-Factors for Cardiovascular Disease in City-Scale Populations

PONE-D-23-34732R1

Dear Dr. Griffiths,

We’re pleased to inform you that your manuscript has been judged scientifically suitable for publication and will be formally accepted for publication once it meets all outstanding technical requirements.

Kind regards,

Keumseok Peter Koh

Academic Editor

PLOS ONE

Additional Editor Comments (optional):

Thank you for your submission and the long waiting for the final decision. It was very challenging to have enough reviewers on the right time since your ABM of CVDs are novel and pioneering. I believe your approach can inspire other ABM modellers for further studies in chronic diseases.

Reviewers' comments:

Reviewer's Responses to Questions

**Comments to the Author**

1. If the authors have adequately addressed your comments raised in a previous round of review and you feel that this manuscript is now acceptable for publication, you may indicate that here to bypass the “Comments to the Author” section, enter your conflict of interest statement in the “Confidential to Editor” section, and submit your "Accept" recommendation.

Reviewer #1: All comments have been addressed

2. Is the manuscript technically sound, and do the data support the conclusions?

Reviewer #1: Yes

3. Has the statistical analysis been performed appropriately and rigorously? 

Reviewer #1: Yes

4. Have the authors made all data underlying the findings in their manuscript fully available?

Reviewer #1: Yes

5. Is the manuscript presented in an intelligible fashion and written in standard English?

Reviewer #1: Yes

6. Review Comments to the Author

Reviewer #1: Thank you for the opportunity to review the

article “An Agent-Based Model of the Spread of

Behavioural risk factors for Cardiovascular

Disease in City-Scale Populations. I greatly appreciate the

efforts of the authors to address all my questions and suggestions.

Regarding the first suggestion, the authors expressed their inability to find the relevant data and discussed it in a discussion section, which gives the readers better clarity.

The second observation was also clarified in the discussion section satisfactorily

The third and fourth observations were also well-addressed.

I appreciate the efforts taken by the authors to thoroughly revise the manuscript.

7. PLOS authors have the option to publish the peer review history of their article (what does this mean?). If published, this will include your full peer review and any attached files.

Reviewer #1: **Yes: **Saju Madavanakadu Devassy

---

## [Editor Report · Acceptance letter]

15 May 2024

PONE-D-23-34732R1 

PLOS ONE

Dear Dr. Griffiths, 

I'm pleased to inform you that your manuscript has been deemed suitable for publication in PLOS ONE. Congratulations! Your manuscript is now being handed over to our production team.

Kind regards, 

on behalf of

Dr. Keumseok Peter Koh 

Academic Editor

PLOS ONE